# The Effect of Fibulin-5 on Hydrocephalus After Subarachnoid Hemorrhage in Mice

**DOI:** 10.3390/ijms26178259

**Published:** 2025-08-26

**Authors:** Yume Suzuki, Mai Nampei, Fumihiro Kawakita, Hiroki Oinaka, Hideki Nakajima, Hidenori Suzuki

**Affiliations:** Department of Neurosurgery, Mie University Graduate School of Medicine, 2-174 Edobashi, Tsu 514-8507, Japan; box_308044@yahoo.co.jp (Y.S.); baske_05@yahoo.co.jp (M.N.); fxmx0216@yahoo.co.jp (F.K.); hiros08290@gmail.com (H.O.); zima0131@gmail.com (H.N.)

**Keywords:** hydrocephalus, extracellular matrix protein, perivascular macrophage, fibulin, subarachnoid hemorrhage

## Abstract

Chronic hydrocephalus following aneurysmal subarachnoid hemorrhage (SAH) is a complication that can lead to deterioration in neurological status and cognitive impairment. Our recent clinical study reported that a high concentration of plasma fibulin-5 (FBLN5), a matricellular protein, was associated with the occurrence of chronic hydrocephalus after SAH. This study aimed to investigate whether and how FBLN5 was associated with hydrocephalus during acute to later phases of SAH in mice. C57BL/6 male mice underwent sham or filament perforation SAH modeling, and vehicle or two dosages (0.01 and 0.1 μg) of short or long recombinant FBLN5 (rFBLN5) were randomly administrated by an intracerebroventricular injection. Neurobehavioral tests, measurements of the degree of ventricular enlargement, Western blotting, and immunohistochemical staining were performed to evaluate hydrocephalus 24 and 48 h after SAH. After SAH, ventricular dilatation did not occur at 24 h but developed at 48 h, and both doses of long rFBLN5 with an arginine–glycine–aspartic acid domain suppressed ventricular dilatation at 48 h after SAH. Long rFBLN5 also decreased phosphorylated p38 in the brain parenchyma and prevented post-SAH increases in perivascular macrophages, as well as microglia activation in the brain parenchyma at 48 h after SAH. Although further research is required to clarify the detailed mechanism, this study demonstrated for the first time that exogenous administration of FBLN5 may have a protective effect against ventricular dilatation after experimental SAH.

## 1. Introduction

Aneurysmal subarachnoid hemorrhage (SAH) accounts for 80% of all cases of SAH [1], representing a condition with poor outcomes and a high mortality rate of 35% [2]. Chronic hydrocephalus following aneurysmal SAH is a complication that can lead to deterioration in neurological status and cognitive impairment, occurring in 9–64% of cases [3,4,5,6,7,8,9,10]. The causes of hydrocephalus include changes in cerebrospinal fluid (CSF) dynamics, obstruction of arachnoid granulations by blood components, and adhesion of the ventricular system [11,12,13]. Research indicates that perivascular macrophage (PVM), one of the border-associated macrophages (BAMs) that plays an important role in the function of the brain boundary tissues and the regulation of the fluid distribution system [14], and leptomeningeal macrophage regulate CSF dynamics [15].

Fibulin-5 (FBLN5) is a matricellular protein (MCP) that is part of the extracellular matrix components [16]. It contains six calcium-binding epidermal growth factor (EGF)-like motifs and an arginine–glycine–aspartic acid (RGD) motif, with a molecular weight of 66-kDa [16]. FBLN5 expression decreases during growth but can re-increase in injured tissues [17,18]. Although prior experimental reports have not established the relationship between SAH and FBLN5, our recent clinical study suggested that elevated plasma levels of FBLN5 in a subacute phase were associated with the subsequent development of chronic hydrocephalus after aneurysmal SAH [19]. As PVM influences CSF dynamics [15], post-SAH changes in PVM may contribute to hydrocephalus development. However, only a few studies have explored the relationship between FBLN5 and macrophages. It was reported that FBLN5 modulates the inflammatory microenvironment, including macrophages in the dermis [20], and that tumor-associated macrophages degrade FBLN5 in epithelial ovarian cancer [21]. This study aimed to investigate whether the administration of recombinant FBLN5 (rFBLN5) influences the development of hydrocephalus following SAH in mice and whether this effect is associated with changes in PVMs.

## 2. Results

### 2.1. Effects of FBLN5 on Ventricular Enlargement at an Acute Phase

In the first experiment, two doses (0.01 and 0.1 μg) of two different lengths of rFBLN5 (short rFBLN5 [SrFBLN5] consisting of only the third and fourth calcium-binding EGF-like motifs and long rFBLN5 [LrFBLN5] that is a nearly full-length rFBLN5 containing the RGD motif) or the vehicle (phosphate-buffered saline [PBS]) were administrated intracerebroventricularly, and the effects on ventricular enlargement were investigated 24 h after SAH by measuring the ventricle-to-brain ratio (VBR), the width of VBR (wVBR), and the area of VBR (aVBR) (Figure 1a and Figure 2). SAH grade and mortality were similar among the SAH groups (Appendix A). Both doses of SrFBLN5 failed to influence post-SAH deterioration of neurological scores (Appendix A), but 0.01 μg of LrFBLN5 administrations improved neurological scores to a degree that was not different from those in the sham group 24 h after SAH (Appendix A). However, neither SAH nor rFBLN5 administration had significant effects on VBR, although higher doses of LrFBLN5 tended to suppress VBR (Appendix A).

### 2.2. Inhibitory Effects of FBLN5 on Ventricular Enlargement at a Later Phase

In the second experiment (Figure 1b), ventricular enlargement was evaluated at 48 h after SAH, because it was impossible to evaluate ventricular enlargement at a chronic phase due to a high mortality of SAH mice. The SAH + vehicle group had a significantly higher mortality rate than the rFBLN5-treated SAH groups, especially the LrFBLN5-treated SAH groups (Appendix A). The SAH + 0.1 μg SrFBLN5 group showed worse neurological scores at 24 and 48 h, but this was thought to reflect its more severe SAH grades (Appendix A). LrFBLN5 administrations did not significantly affect SAH grades and neurological scores, but a higher dose of LrFBLN5 tended to improve neurological scores at 48 h (Appendix A). As to ventricular enlargement at 48 h, both wVBR and aVBR were significantly higher in the SAH + vehicle group than the sham + vehicle group, and LrFBLN5 administration improved them to a level not significantly different from those in the sham group, especially in terms of wVBR (Figure 3).

### 2.3. Exploring Mechanisms for LrFBLN5 to Prevent Post-SAH Ventricular Dilatation

Western blotting (WB) analyses were performed to investigate the possible mechanisms for LrFBLN5 to inhibit ventricular dilatation and therefore changes in expressions of proteins potentially affected by LrFBLN5 (Figure 1c). There were no significant differences in the mortality and SAH grades among the SAH groups (Appendix A). Neurological scores were similar to the findings in experiment 2 (Appendix A). WB showed that SAH did not significantly increase any of the proteins tested including transforming growth factor (TGF)-β1, Smad, mitogen-activated protein kinases (MAPKs), and tenascin-C (TNC). However, phosphorylated p38 was markedly decreased in the SAH + LrFBLN5 groups compared with the SAH + vehicle group (Figure 4).

### 2.4. Effects of LrFBLN5 on Ionized Calcium-Binding Adaptor Molecule 1 (Iba1)-Positive Cells in the Brain Parenchyma and Perivascular Space

Next, immunohistochemical staining of Iba1 was performed to assess the effect of LrFBLN5 on PVMs after SAH (Figure 1d). The SAH grade was unexpectedly more severe in the higher dose group, but the mortality and neurological scores were similar to those in experiments 2 and 3 (Figure 5a–c and Appendix A). Iba1-positive cells were significantly increased in both the brain parenchyma and the perivascular space in the SAH + vehicle group compared with the sham + vehicle group and were suppressed by the administration of LrFBLN5 (Figure 5d,e).

## 3. Discussion

The novel findings in this study were as follows: (1) administration of LrFBLN5 tended to improve neurobehavioral functions in a dose-independent manner; (2) ventricular dilatation did not occur at 24 h, but developed 48 h after SAH; (3) both doses of LrFBLN5 suppressed ventricular dilatation 48 h after SAH; and (4) LrFBLN5 decreased phosphorylated p38 in the brain parenchyma and prevented post-SAH increases in activated microglia in the brain parenchyma and PVMs 48 h after SAH.

The causes of hydrocephalus include changes in CSF dynamics, fibrosis of the leptomeninges and arachnoid granulations due to blood components, and adhesion in the ventricular system [11,12,13,22,23]. Previous research indicated that PVMs and leptomeningeal macrophages regulate CSF dynamics [15].

PVMs are classified as BAMs, which also encompass meningeal macrophages and choroid plexus macrophages [24]. BAMs and microglia play critical roles in the brain’s innate immunity and inflammation [24]. The functions of PVM include perivascular drainage, cerebrovascular flexibility, phagocytic activity, antigen presentation, inflammatory responses, and maintenance of blood–brain barrier integrity [25]. Under normal conditions, PVM functions as a scavenger and surveillance cell; however, in pathological states, it can have detrimental effects [24]. Following SAH, PVM interacts with erythrocytes and other blood components, increasing perivascular inflammation and contributing to microvascular thrombus formation [26,27,28]. PVM also participates in neuronal apoptosis and perivascular gliosis after SAH [28]. Experimental models showed that depleting of PVM with clodronate could suppress perivascular inflammation, neuronal apoptosis, and perivascular gliosis, improving the outcomes after SAH [28].

PVMs are integral to the glymphatic system [29]. CSF enters the perivascular space, penetrating the arterial walls and flowing into the interstitial space of the brain [29]. PVMs function as gatekeepers, preventing the accumulation of large particles in the perivascular space and reducing the inhibition of glymphatic flux [28,30]. They also regulate the flow rate of CSF by inducing contraction and relaxation in vascular smooth muscle cells [15]. However, under SAH, PVMs are thought to cause perivascular inflammation, thereby restricting arteriolar pulsation and compromising CSF flow [26,27,28,30]; therefore, it is possible that the clearance of CSF may improve via PVM depletion from the glymphatic system after SAH.

FBLN5 is a 66-kDa MCP that plays a crucial role in organizing elastic fibers and mediating various cellular functions essential for tissue development and homeostasis [16,31,32]. It consists of six calcium-binding EGF-like motifs and fibulin modules, including one motif containing the RGD sequence [16]. FBLN5 expression decreases during growth but increases again in injured tissues [17,18]. Our previous research suggested that in clinical settings, the elevated plasma levels of FBLN5 during the subacute phase of aneurysmal SAH may contribute to the development of chronic hydrocephalus [19]. Considering the results of previous studies and the present experiments, it is possible that an increase in plasma FBLN5 may have a protective effect against hydrocephalus; however, due to the progressive natural course of the disease, it is likely that the naturally increased amount of FBLN5 is not enough to prevent the subsequent onset of hydrocephalus or to fully reverse hydrocephalus at that point.

Only a few studies have explored the relationship between FBLN5 and macrophages. FBLN5 is involved in regulating the inflammatory microenvironment in the dermis of Snail transgenic mice and influences the proliferation and reduction of macrophages [20]. In epithelial ovarian cancer, FBLN5 is degraded by proteases prevalent in the tumor microenvironment macrophages, with the degraded form facilitating ovarian cancer cell adhesion and local metastasis [21]. In a mouse spinal cord injury model, small extracellular vesicles derived from CD163 macrophages, which were the same as PVM, were shown to promote vascular regeneration and stabilization when modified with the RGD motif [33], suggesting the possibility of an interaction between the RGD, including FBLN5 and PVM. Our experimental results suggested that PVMs, which may adversely affect CSF dynamics in the perivascular space following SAH, were depleted by LrFBLN5 administration, leading to the improvement of post-SAH ventricular enlargement at 48 h (Figure 6). The observed improvement in neurological outcomes was not necessarily consistent with the inhibitory effect related to ventricular enlargement. This discrepancy can be explained by the fact that LrFBLN5 was also protective against early brain injury after experimental SAH [34]. The authors believe that further insight into these effects can be gained by evaluating the effects of LrFBLN5 on ventricular enlargement and neurological outcome in the chronic phase.

Following SAH, TGF-β1 levels in the CSF increased, suggesting its involvement in subarachnoid fibrosis and chronic hydrocephalus [35,36,37,38]. Additionally, it is proposed that Smad proteins and MAPKs, which relate to this pathway, may also play a role in fibrosis [39,40,41,42,43]. TNC, a type of MCP, is implicated in neuronal apoptosis, breakdown of the blood–brain barrier, and vasospasm following SAH [44,45]. TNC can also promote tissue fibrosis by enhancing leptomeningeal collagen synthesis, leading to ventricular enlargement and the development of chronic hydrocephalus [45,46]. Similar to TGF-β1, TNC is associated with various inflammatory cytokines such as interleukins [47,48,49]. In this study, it is unclear why these proteins did not increase significantly after SAH in the WB. The possible explanations include the following: (1) although they increased at the meningeal level, their expression changes were not detected well in the evaluation of the entire brain; and (2) because this study was conducted 48 h after SAH, it was too early to detect changes in their proteins’ expression, which could have been captured more clearly if they had been evaluated in the chronic phase. However, this study revealed that the suppression of p38 activation by LrFBLN5 may have been involved in its preventive effects against the activation of PVM and microglia, as well as ventricular enlargement after SAH, although further detailed investigation is required.

The present study has several limitations. First, all experiments used only male mice because this study was intended to clarify the pathophysiology rather than therapeutic intent, and differences in the immune system between the sexes may have affected the results. Second, experiments were conducted only within 48 h of SAH and investigated acute rather than chronic hydrocephalus. We also attempted to investigate mice 1 week after modeling, but the survival rates were low: 3 out of 5 in the SAH + vehicle group, 1 out of 11 in the SAH + 0.01 μg SrFBLN5 group, and 0 out of 6 in the SAH + 0.1 μg SrFBLN5 group. Although this limitation arose from the difficulty in achieving long-term survival in SAH + vehicle and SAH + SrFBLN5 mice, further evaluations during the chronic phase, focusing specifically on the long-term effects of LrFBLN5, are needed. In the mouse endovascular perforation model, a higher mortality rate has been reported compared with the cisterna magna injection model or the chiasmatic cistern injection model [50], indicating the need for another model to examine later phases. Third, no experiments verified the mechanism by which LrFBLN5 affected PVM and ventricular enlargement, although the difference in the effects of LrFBLN5 and SrFBLN5 suggested the involvement of the RGD motif. Fourth, WB analysis was performed on whole brains, and therefore it may not have been possible to accurately detect changes in locally expressed proteins. Further experiments are needed to clarify the mechanistic relationships between FBLN5, PVM, and p38 signaling pathways. However, this is the first study to experimentally show the relationship between FBLN5, PVM, and hydrocephalus.

## 4. Materials and Methods

All procedures were approved by the Animal Ethics Review Committee of Mie University and were carried out according to the institution’s and the Animals in Research: Reporting In Vivo Experiments (ARRIVE) guidelines. After SAH injection, animals were randomly assigned to one of the treatment or control groups by drawing lots. Data collection and analyses were performed by a researcher blinded to the treatment group.

### 4.1. Study Protocols (Figure 1)

First, to investigate the effect of FBLN5 on ventricular enlargement in an acute phase after SAH, 59 mice were randomly divided into 6 groups: sham + vehicle (PBS) (*n* = 12), SAH + vehicle (*n* = 16), SAH + low-dose (0.01 μg) SrFBLN5 (*n* = 8), SAH + high-dose (0.1 μg) SrFBLN5 (*n* = 7), SAH + low-dose LrFBLN5 (*n* = 9), and SAH + high-dose LrFBLN5 (*n* = 7) (Figure 1a and Figure 2a). Based on our preliminary study [34], two doses of SrFBLN5 or LrFBLN5 and the vehicle were administered intracerebroventricularly at 30 min post-modeling. After neurobehavioral function was assessed 24 h after modeling, mice were euthanized to evaluate SAH grades, wVBR, and aVBR, as shown in Figure 2b, which were separately compared between the vehicle-treated and two doses of SrFBLN5-treated groups, and among the vehicle-treated and two doses of LrFBLN5-treated groups.

Second, to investigate the effect of FBLN5 on ventricular enlargement in a later phase after SAH, 100 mice were randomly divided into 6 groups: sham + vehicle (*n* = 16), SAH + vehicle (*n* = 39), SAH + low-dose SrFBLN5 (*n* = 15), SAH + high-dose SrFBLN5 (*n* = 14), SAH + low-dose LrFBLN5 (*n* = 7), and SAH + high-dose LrFBLN5 (*n* = 9) (Figure 1b). Drugs were administered via intracerebroventricular infusion (ICV) as described above. After neurobehavioral tests were performed 24 and 48 h after modeling, mice were euthanized, and SAH grade, wVBR, and aVBR were measured. Comparisons were made separately between the vehicle-treated and 2 doses of SrFBLN5-treated groups and among the vehicle-treated and 2 doses of LrFBLN5-treated groups.

Third, to elucidate the mechanisms of LrFBLN5’s inhibitory effects on ventricular enlargement, 29 mice were randomly divided into 4 groups: sham + vehicle (*n* = 6), SAH + vehicle (*n* = 8), SAH + low-dose LrFBLN5 (*n* = 6), and SAH + high-dose LrFBLN5 (*n* = 9) (Figure 1c). Drug administration was performed via ICV as described above. After neurobehavioral tests were assessed 24 and 48 h after modeling, mice were euthanized, and SAH grading and WB were performed.

Fourth, to assess the effect of LrFBLN5 on macrophages in the perivascular space, 37 mice were randomly divided into 4 groups: sham + vehicle (*n* = 6), SAH + vehicle (*n* = 15), SAH + low-dose LrFBLN5 (*n* = 7), and SAH + high-dose LrFBLN5 (*n* = 9) (Figure 1d). Drug administration was the same as described above. After neurobehavioral tests were assessed 24 and 48 h after modeling, mice were euthanized to perform SAH grading and immunohistochemical staining.

### 4.2. rFBLN5 (Figure 2a)

To clarify the functional site of FBLN5, two different lengths of rFBLN5 were administrated intracerebroventricularly: a SrFBLN5 consisting of only the third and fourth calcium-binding EGF-like motifs (RPD153Mu02, Cloud-Clone Corp, Houston, TX, USA) and a nearly full-length rFBLN5 containing the RGD motif (LrFBLN5; 9006-FB, R&D System, Minneapolis, MN, USA).

### 4.3. SAH Modeling

The sample size to test the primary outcome (neurological deterioration) was determined based on calculations using our preliminary findings (a power of 0.8, effect size η^2^ of 0.404–0.676, and alpha of 0.05; statistical package for social science version 28, IBM Corp., Armonk, NY, USA; *n* = 3–6/group) and previous studies using the same SAH models (*n* = 6–10/group) [51]. C57BL/6 male adult mice (age 10–12 weeks, 25–30 g; SLC, Hamamatsu, Japan) were used for this study. As this study was intended to clarify the pathophysiology rather than therapeutic intent, we used only male mice. Experiments were performed at an animal testing laboratory. Mice were housed in cages for at least 24 h after transport to the laboratory and then underwent surgery. As previously described, mice underwent endovascular perforation SAH or sham modeling [51]. Mice were anesthetized with an intraperitoneal injection of mixed 3-type anesthetic agents (0.75 mg/kg of medetomidine hydrochloride (Fujita Pharm, Tokyo, Japan), 4 mg/kg of midazolam (Sandoz, Tokyo, Japan), and 5 mg/kg of butorphanol tartrate (Meiji Animal Health, Tokyo, Japan)). After the anesthesia, mice were placed in a supine position, and a skin incision was made at the midline of the neck to expose the left carotid arteries. A 4-0 nylon monofilament with a sharpened tip was inserted from the left external carotid artery stump into the left internal carotid artery about 15 mm to perforate the bifurcation of the left anterior cerebral artery and the left middle cerebral artery. Then, the filament was withdrawn, and the stump of external carotid artery was coagulated. The wound was sutured. The sham mice underwent the same procedure as described above, except that the artery was not perforated. During the operation, blood pressure and heart rate were monitored via the tail, and body temperature was kept at 37 °C.

### 4.4. ICV

At 30 min post-modeling, surviving mice underwent ICV under the continued sedation as previously described [51,52]. Mice were placed in a stereotactic head holder, and a skin incision was made at the midline of the head. The needle of a 2 μL Hamilton syringe (Hamilton Company, Reno, NV, USA) was inserted via the burr hole perforated on the skull into the left lateral ventricle using the following coordinates relative to the bregma: 0.2 mm posterior, 1.0 mm lateral, and 2.25 mm below the horizontal plane of the bregma. Sterile 2 μL vehicle (PBS, regulated to pH 7.2–7.4) with and without SrFBLN5 or LrFBLN5 (0.01 or 0.1 μg) was injected at a rate of 1 μL/min. The needle was gently removed 5 min after an injection, and the wound was quickly sutured. After surgery, mice were returned to clean cages and allowed free access to food and water, and the room temperature was kept constant at 25 ± 1 °C.

### 4.5. Neurobehavioral Test

Neurobehavior functions were blindly assessed using the modified Garcia’s neurological score system as previously described [52,53]. The evaluation consisted of six tests scored 0 to 3 or 1 to 3. The six tests included spontaneous activity, spontaneous movement of four limbs, forepaw outstretching, climbing, body proprioception, and response to whisker stimulation. Mice were given a score of 2 to 18 in 1-number steps, and higher scores indicated better function.

### 4.6. SAH Grade and Exclusion Criteria

The severity of SAH was blindly evaluated using high-resolution pictures of the base of the brain taken at each sacrifice. Two evaluators (Y.S. and M.N.) scored each model and the average of these scores was calculated. The SAH grading system was as follows. The basal cistern was divided into six segments, and each segment was allotted a grade from 0 to 3 depending on the amount of subarachnoid blood clot in the segment: grade 0, no subarachnoid blood; grade 1, minimal subarachnoid blood; grade 2, moderate blood clot with recognizable arteries; and grade 3, blood clot obliterating all arteries within the segment [53]. The mice received a total score ranging from 0 to 18 after adding the scores from all six segments. Mice with SAH grading scores ≤7 at 24 h and ≤4 at 48 h were excluded because they had no significant brain injury according to our preliminary study.

### 4.7. VBR (Figure 2b)

The degree of ventricular enlargement was blindly and morphologically evaluated using the coronal section 0.5 mm anterior to bregma as previously described [54]. Mice were deeply anesthetized and transcardially perfused with 30 mL PBS followed by 15 min of 10% neutral buffered formalin at 60–80 mmHg. Brains were fixed in 10% neutral buffered formalin for approximately 12 h and embedded in paraffin. Then, 4 µm thick coronal sections at 0.5 mm anterior to the bregma were cut and mounted on the slide. The width and area were quantified by densitometric analyses using Image J Version 1.54p software (NIH, Bethesda, MD, USA). The percentage of VBR was calculated in two ways according to the following formula: wVBR = (maximum width of the right lateral ventricle + maximum width of the left lateral ventricle)/maximum width of the brain, which was previously used to evaluate hydrocephalus [54]; and aVBR = (area of the right lateral ventricle + area of the left lateral ventricle)/area of the brain.

### 4.8. WB

WB was performed as previously described [52]. The left cerebral hemisphere was used for analyses. Equal amounts of protein samples were separately loaded on SDS-PAGE gels, electrophoresed, and transferred onto a polyvinylidene difluoride membrane. The membranes were blocked with 5% bovine serum albumin or 5% *w*/*v* nonfat dry milk followed by incubation overnight at 4 °C with the following primary antibodies: mouse monoclonal TGF-β1 (1:200, sc-52893; Santa Cruz Biotechnology, Santa Cruz, CA, USA), rabbit monoclonal anti-Smad 2/3 (1:1000, ab202445; Abcam, Cambridge, UK), rabbit monoclonal anti-phosphorylated Smad 2/3 (1:1000, ab254407; Abcam, Cambridge, UK), rabbit monoclonal anti-phosphorylated p38 (1:1000, #4511; Cell Signaling Technology, Danvers, MA, USA), mouse monoclonal anti-phosphorylated c-Jun N-terminal kinase (JNK; 1:1000, sc-6254; Santa Cruz Biotechnology, Santa Cruz, CA, USA), rabbit monoclonal anti-phosphorylated extracellular signal-related kinase (ERK) 1/2 (1:1000, #4370; Cell Signaling Technology, Danvers, MA, USA), rabbit monoclonal anti-ERK1/2 (1:1000, #4695; Cell Signaling Technology, Danvers, MA, USA), and rabbit monoclonal anti-TNC (1:1000, ab108930; Abcam, Cambridge, UK) antibodies. Then, the membrane was incubated with goat anti-rabbit secondary antibodies (PI-1000; Vector, Burlingame, CA, USA) or anti-mouse secondary antibodies (PI-2000; Vector, Burlingame, CA, USA) for 1 h at room temperature. A chemiluminescence reagent kit (ECL Prime; Amersham Bioscience, Arlington Heights, IL, USA) was used to detect immunoreactive bands. The bands were quantified by densitometric analyses using Image J Version 1.54p software (NIH, Bethesda, MD, USA). β-tubulin (1:2000, #2146; Cell Signaling Technology, Danvers, MA, USA) was used as a loading control.

### 4.9. Immunohistochemical Staining

Immunohistochemical staining was performed as previously described [52]. Mice were deeply anesthetized and transcardially perfused with 30 mL PBS followed by 15 min of 10% neutral buffered formalin at 60–80 mmHg. Brains were fixed in 10% neutral buffered formalin for approximately 12 h and embedded in paraffin. Then, 4 µm thick coronal sections at 0.5 mm anterior to the bregma were cut and mounted on the slide. After the sections were dewaxed and dehydrated, antigen retrieval was performed in 1 mmol/L ethylenediaminetetraacetic acid (pH 8.0) at 80 °C for 20 min. To quench any endogenous peroxidase activity, the sections were incubated in 3% hydrogen peroxide for 10 min, followed by being blocked with normal serum for 60 min at room temperature. Then, the sections were incubated with rabbit monoclonal anti-Iba1 (1:2000, ab178847; Abcam, Cambridge, UK) antibody as the primary antibody overnight at 4 °C, followed by incubation with biotinylated goat anti-rabbit polyclonal immunoglobulin G (1:200; Vector Laboratories, Burlingame, CA, USA) as the secondary antibody for 30 min at room temperature. Sections were then incubated with an avidin–biotin–horseradish peroxide complex (Vectastain ABC Kit; Vector Laboratories, Burlingame, CA, USA) for 30 min at room temperature. The sections were visualized by diaminobenzidine/hydrogen peroxide solution and counterstained with hematoxylin for light microscopic examination.

To evaluate the expression of Iba1, four continuous pictures of the left (perforation side) secondary somatosensory cortex [55,56] at ×200 magnification were photographed under a light microscope. The relative quantity of Iba1 per picture was measured by integrated optical density using Image Pro Plus 6.0 software (Media Cybernetics Inc., Rockville, MD, USA), and the average value in the four continuous pictures was calculated.

### 4.10. Statistical Analysis

All statistical analyses were performed with SPSS software, version 30.0 (IBM, Armonk, NY, USA). After normality testing with the Shapiro–Wilk test, continuous variables were presented on graphs as means ± standard error of the mean and compared among three or more groups using the Kruskal–Wallis test followed by post hoc Steel–Dwass multiple comparisons or one-way analysis of variance (ANOVA) followed by post hoc Tukey multiple comparisons, as appropriate. Rank scales were presented on graphs as means ± standard error of the mean and tested using Kruskal–Wallis tests followed by post hoc Steel–Dwass multiple comparisons. Mortality was analyzed using Fisher’s exact test. A significant level was set at a *p* value < 0.05.

## 5. Conclusions

This study first revealed that exogenous administration of FBLN5 may have a protective effect against ventricular dilatation after experimental SAH.

## Figures and Tables

**Figure 1 ijms-26-08259-f001:**
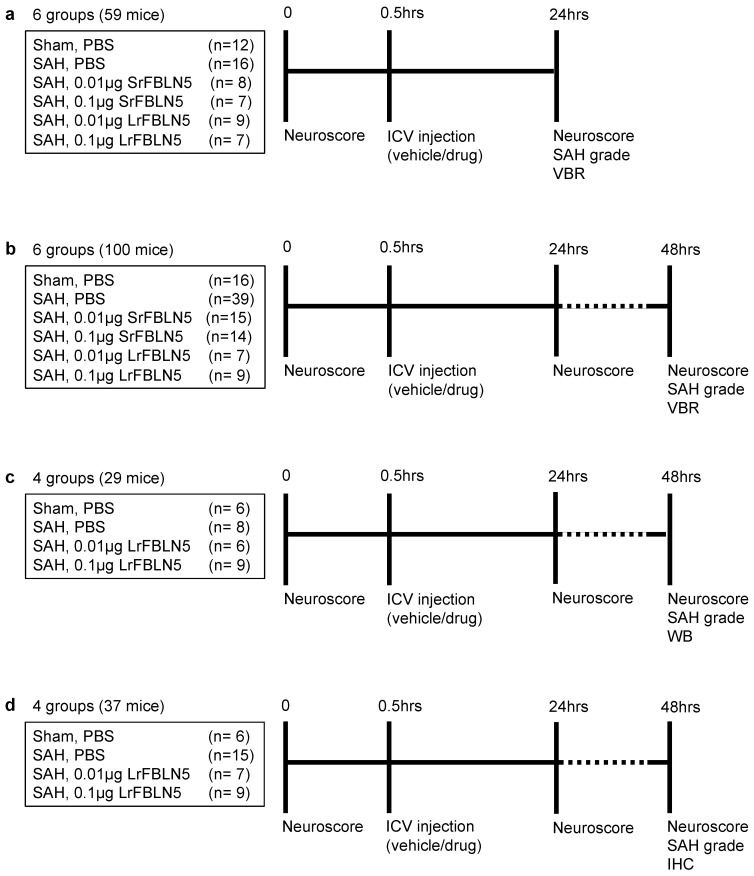
Experimental designs. Experiment 1 to evaluate effects of fibulin-5 (FBLN5) against ventricular dilatation 24 h after subarachnoid hemorrhage (SAH) (**a**); experiment 2 to evaluate effects of FBLN5 against ventricular dilatation 48 h after SAH (**b**); experiment 3 to elucidate the mechanisms of FBLN5’s effect on hydrocephalus 48 h after SAH (**c**); and experiment 4 to evaluate effects of FBLN5 against activation of microglia and macrophage in the brain parenchyma 48 h after SAH (**d**). ICV—intracerebroventricular; IHC—immunohistochemistry; LrFBLN5—long recombinant FBLN5; PBS—phosphate-buffered saline; SrFBLN5—short recombinant FBLN5; VBR—ventricular-to-brain ratio; WB—Western blot.

**Figure 2 ijms-26-08259-f002:**
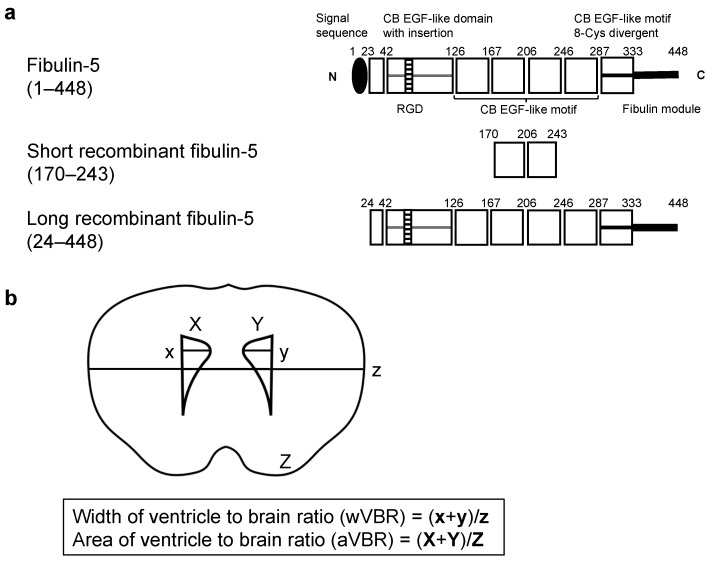
The structure schemas of fibulin-5 (FBLN5) (**a**) and the calculation method of the ventricular dilatation (**b**). Full-length FBLN5 (upper), the short type of recombinant FBLN5 (middle), and the long type of recombinant FBLN5 (lower). CB EGF—calcium-binding epidermal growth factor; RGD—arginine–glycine–aspartic acid.

**Figure 3 ijms-26-08259-f003:**
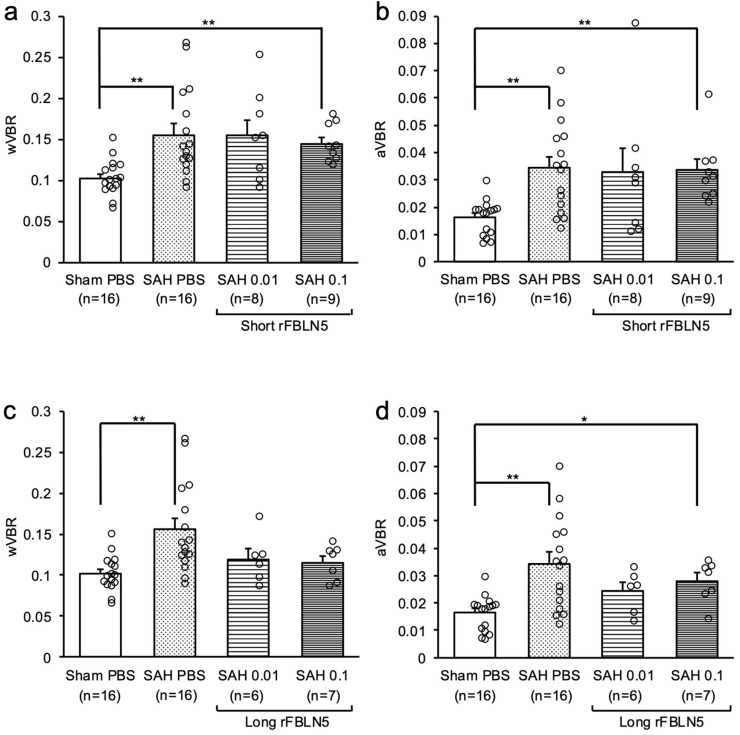
The effects of two dosages (0.01 or 0.1 μg) of administered short or long recombinant fibulin-5 (rFBLN5) on ventricle-to-brain ratio 48 h after subarachnoid hemorrhage (SAH). * *p* < 0.05 vs. sham phosphate-buffered saline (PBS) group, and ** *p* < 0.01 vs. sham PBS group; Kruskal–Wallis test (**a**,**b**,**d**) or one-way ANOVA (**c**). aVBR—area of ventricle-to-brain ratio; wVBR—width of ventricle-to-brain ratio.

**Figure 4 ijms-26-08259-f004:**
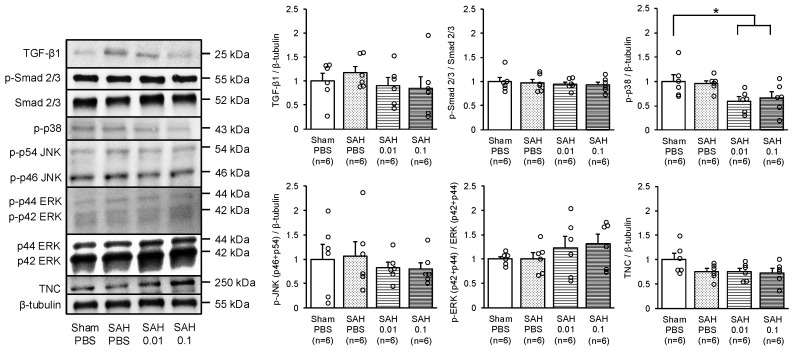
The effects of two dosages (0.01 or 0.1 μg) of administered long recombinant fibulin-5 on expressions of transforming growth factor (TGF)-β1, Smad 2/3, p38, c-Jun N-terminal kinase (JNK), extracellular signal-related kinase (ERK) 1/2, and tenascin-C (TNC) in the left cerebral hemisphere 48 h after subarachnoid hemorrhage (SAH). * *p* < 0.05 vs. sham phosphate-buffered saline (PBS) group; one-way ANOVA. p-—phosphorylated.

**Figure 5 ijms-26-08259-f005:**
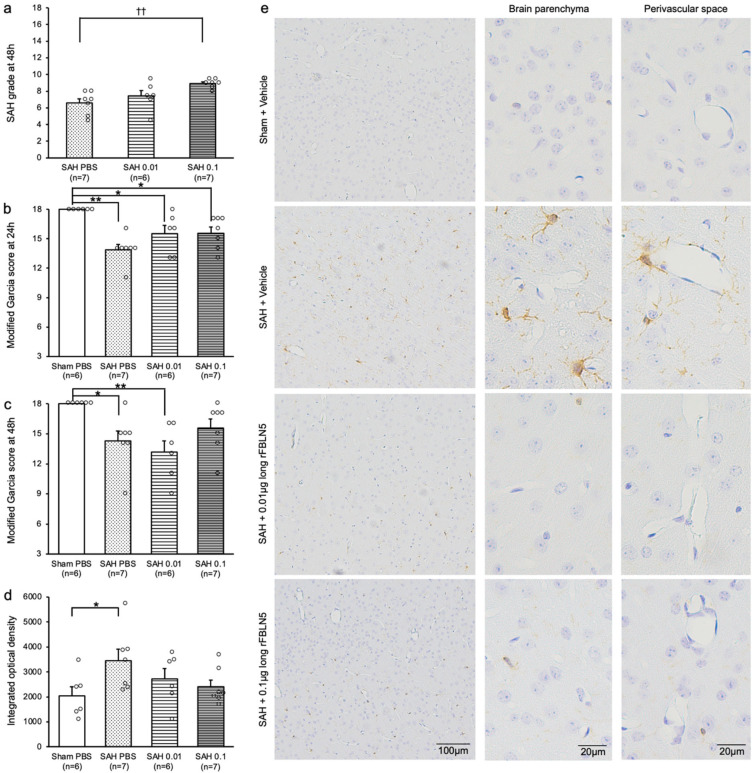
The effects of 2 dosages (0.01 or 0.1 μg) of long recombinant fibulin-5 (rFBLN5) administration on subarachnoid hemorrhage (SAH) grade (**a**), neurological score at 24 h (**b**) and 48 h (**c**), and the ionized calcium binding adaptor molecule 1 staining in brain parenchyma (integrated optical density (**d**) and representative images (**e**)) 48 h after SAH. * *p* < 0.05 vs. sham phosphate-buffered saline (PBS) group, ** *p* < 0.01 vs. sham PBS group, and ^††^ *p* < 0.01 vs. SAH PBS group; Kruskal–Wallis test (**a**–**c**) or ANOVA (**d**).

**Figure 6 ijms-26-08259-f006:**
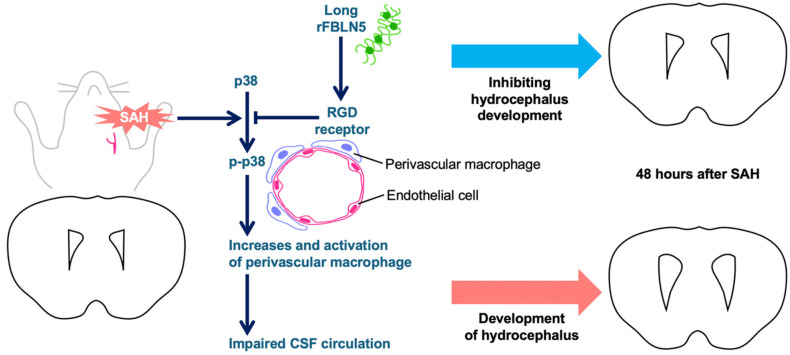
The schema of the hypothesis regarding the relationship between fibulin-5, perivascular macrophages, and hydrocephalus in a mouse model of subarachnoid hemorrhage (SAH). CSF—cerebrospinal fluid; rFBLN5—recombinant fibulin-5; RGD—arginine–glycine–aspartic acid.

## Data Availability

Data from this study are available to qualified investigators upon reasonable inquiry.

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
