# Peer review of "The Effect of Fibulin-5 on Hydrocephalus After Subarachnoid Hemorrhage in Mice"

_ijms, 2025, doi:10.3390/ijms26178259_

Round 1

Reviewer 1 Report

Comments and Suggestions for Authors

The authors have done a very nice job of designing and presenting their data in this rather novel look at the possible effects of fibulin 5 on the development of hydrocephalus. There is a tremendous amount of data and this is a very long paper.

The results are very nicely presented. The key figure is the immunohistochemistry that shows the staining in the vehicle mice vs the anti-fibulin mice.

Here are a couple of comments/clarifications that may help the readership:

  1. Do you have any data with this model as to what happens to the mice after 48 hours? Do they survive or do they all die?
  2. If the treated mice all die, that would suggest that the effect of the anti-fibulin treatment is transient and that the inflammatory process that started with the SAH continues or may have a different time course than anticipated
  3. The model is more that of an acute hydrocephalus rather than chronic. 

Author Response

Comment 1: Do you have any data with this model as to what happens to the mice after 48 hours? Do they survive or do they all die?

Response 1: Thank you for your suggestions. When the survival rate was investigated one week after modeling in mice other than those used in this experimental system, 3 out of 5 mice in the SAH-vehicle group, 1 out of 11 mice in the SAH-SrFBLN5 0.01μg group, and 0 out of 6 mice in the SAH-SrFBLN5 0.1μg group were confirmed to have survived. However, the survival rate one week after administration was not investigated in the SAH-LrFBLN5 group. According to your suggestions, we added the following sentences on lines 215 to 217: “We also attempted to investigate mice 1 week after modeling, but the survival rates were low: 3 out of 5 in the SAH + vehicle group, 1 out of 11 in the SAH + 0.01μg SrFBLN5 group, and 0 out of 6 in the SAH + 0.1μg SrFBLN5 group.”

Comment 2: If the treated mice all die, that would suggest that the effect of the anti-fibulin treatment is transient and that the inflammatory process that started with the SAH continues or may have a different time course than anticipated

Response 2: Thank you for your suggestions. We have not investigated whether LrFBLN5 administration affects the survival rate of SAH mice, so it is unknown. However, we agree that the long-term effects need to be verified. Based on your suggestions, we have revised lines 217 to 220 as follows: “Although this limitation arose from the difficulty in achieving long-term survival in SAH + vehicle and SAH + SrFBLN5 mice, further evaluations during the chronic phase, focusing specifically on the long-term effects of LrFBLN5, are needed.”

Comment 3: The model is more that of an acute hydrocephalus rather than chronic.

Response 3: Thank you for your suggestions. We agree with you. According to your suggestions, we specified this as one of limitations on lines 213 to 215 as follows: “Second, experiments were conducted only within 48 hours of SAH and investigated acute rather than chronic hydrocephalus.”

Reviewer 2 Report

Comments and Suggestions for Authors

The manuscript “The Effect of Fibulin-5 on Hydrocephalus After Subarachnoid Hemorrhage in Mice,” presents a well-designed and timely study investigating the therapeutic potential of recombinant fibulin-5 (FBLN5) in post-SAH hydrocephalus. The authors translate prior clinical observations into a mechanistic mouse model, demonstrating that long recombinant FBLN5 (LrFBLN5), containing the RGD motif, suppresses ventricular dilatation, reduces phosphorylated p38 signaling, and attenuates perivascular macrophage and microglial activation at 48 hours post-SAH. These findings provide novel insight into extracellular matrix protein involvement in post-SAH pathology.

The study is robust, employing a clinically relevant endovascular perforation model, multiple experimental groups, and dose/length comparisons of recombinant FBLN5. Methodological rigor, including blinded assessments and adherence to ARRIVE guidelines, enhances the reliability of the results. The inclusion of both short and long forms of FBLN5 strengthens the conclusion that the RGD motif is critical to its protective effect. The discussion is thoughtful, addressing both mechanistic implications and the paradox between exogenous versus endogenous FBLN5 levels.

Some limitations should be noted.

  • Analyses were restricted to 48 hours post-SAH due to model mortality, limiting conclusions about chronic hydrocephalus.
  • Only male mice were studied, raising questions about sex differences.
  • Western blot analyses on whole brain tissue may obscure region-specific changes, and the mechanistic link between FBLN5, perivascular macrophages, and p38 signaling requires further clarification.
  • The partial inconsistency between ventricular protection and neurobehavioral outcomes deserves additional exploration.

Overall, this is a strong and original contribution that advances our understanding of FBLN5 in SAH-related hydrocephalus and identifies it as a potential therapeutic target. With minor revisions addressing the points above and ensuring clear figure presentation, the manuscript is suitable for publication.

Author Response

Some limitations should be noted.

Comment 1: Analyses were restricted to 48 hours post-SAH due to model mortality, limiting conclusions about chronic hydrocephalus.

Response 1: Thank you for your suggestions. According to your suggestions, we specified this as one of limitations on lines 213 to 223 as follows: “Second, experiments were conducted only within 48 hours of SAH and investigated acute rather than chronic hydrocephalus. We also attempted to investigate mice 1 week after modeling, but the survival rates were low: 3 out of 5 in the SAH + vehicle group, 1 out of 11 in the SAH + 0.01μg SrFBLN5 group, and 0 out of 6 in the SAH + 0.1μg SrFBLN5 group. Although this limitation arose from the difficulty in achieving long-term survival in SAH + vehicle and SAH + SrFBLN5 mice, further evaluations during the chronic phase, focusing specifically on the long-term effects of LrFBLN5, are needed. In the mouse endovascular perforation model, a higher mortality rate has been reported compared to the cisterna magna injection model or the chiasmatic cistern injection model [51], indicating the need for another model to examine later phases.”

Comment 2: Only male mice were studied, raising questions about sex differences.

Response 2: Thank you for your suggestions. However, this point has already been listed on lines 211 to 213 as one of the limitations as follows: “First, all experiments used only male mice because this study was intended to clarify the pathophysiology rather than therapeutic intent, and differences in the immune system between the sexes may have affected the results.”

Comment 3: Western blot analyses on whole brain tissue may obscure region-specific changes, and the mechanistic link between FBLN5, perivascular macrophages, and p38 signaling requires further clarification.

Response 3: Thank you for your suggestions. According to your suggestions, we added the following sentence to the limitations section on lines 227 to 228: “Further experiments are needed to clarify the mechanistic relationships between FBLN5, PVM, and p38 signaling pathways.”

Comment 4: The partial inconsistency between ventricular protection and neurobehavioral outcomes deserves additional exploration.

Response 4: Thank you for your suggestions. As described on lines 189 to 192, our previous reports suggested that FBLN5 also has protective effects against early brain injury within 48 hours after SAH. The authors believe that further insight into these effects can be gained by evaluating the effects of FBLN5 on ventricular enlargement and neurological outcome in the chronic phase. According to your suggestions, we added this to lines 192 to 194.